# Liposomes Encapsulating Morin: Investigation of Physicochemical Properties, Dermal Absorption Improvement and Anti-Aging Activity in PM-Induced Keratinocytes

**DOI:** 10.3390/antiox11061183

**Published:** 2022-06-16

**Authors:** Hong-My Tran, Chun-Yin Yang, Tzu-Hui Wu, Feng-Lin Yen

**Affiliations:** 1Department of Fragrance and Cosmetic Science, College of Pharmacy, Kaohsiung Medical University, Kaohsiung 807, Taiwan; u109533011@kmu.edu.tw; 2School of Pharmacy, College of Pharmacy, Kaohsiung Medical University, Kaohsiung 807, Taiwan; anitayang0812@gmail.com; 3Department of Pharmacy and Master Program, Tajen University, Pingtung County 90741, Taiwan; 4Drug Development and Value Creation Research Center, Kaohsiung Medical University, Kaohsiung 807, Taiwan; 5Department of Medical Research, Kaohsiung Medical University Hospital, Kaohsiung 807, Taiwan; 6Institute of Biomedical Sciences, National Sun Yat-sen University, Kaohsiung 807, Taiwan

**Keywords:** morin, liposomes, water solubility, skin penetration, PM, skin anti-aging

## Abstract

Recently, a global market for anti-aging skin care using botanicals has been noticeably developing. Morin, 3,5,7,2′,4′-pentahydroxyflavone, is a polyphenol with many pharmacological properties including antioxidant, anti-inflammation and photoprotection. However, poor aqueous solubility of morin restricts its application in pharmaceuticals. The present study aimed to encapsulate morin into liposomal vesicles to improve its water solubility and skin penetration, and further investigated its ROS inhibition and anti-aging activity in HaCaT keratinocytes induced by particulate matters (PMs). Our data presented that morin was a strong DPPH^•^ radical scavenger. Morin displayed a remarkable ROS inhibitory ability and protected keratinocytes against PMs by downregulating matrix metalloproteinase-1 (MMP-1) expression via suppressing p-ERK and p-p38 in the MAPK pathway. Moreover, water solubility of liposomal morin (LM) prepared by the thin film hydration method was significantly better than free form of morin due to particle size reduction of LM. Our results also demonstrated that deformable liposomal vesicles were achieved for increasing dermal absorption. Additionally, LM (morin:lecinolws-50:tween-80:PF-68, 1:2.5:2.5:5) was able to effectively reduce generation of ROS, inactivate p-ERK, p-p38 and MMP-1 in HaCaT cells exposed to PM. In conclusion, our findings suggested that LM would be a bright candidate for various topical anti-aging and anti-pollution products.

## 1. Introduction

Despite COVID-19 pandemic crisis, global anti-aging market size is expected to keep booming to reach USD 83.2 billion by 2027, equivalent to a growth rate of 6.8% (Research and Market, 2020). Similarly, personal care products using botanical extracts were projected to see a rise of 10–11% annually from 2015 to 2019 and are expected to explode in the coming years [1]. Among numerous plant-derived compounds, polyphenols are the most commonly found active ingredients for cosmetic and skin care application with many benefits [2,3]. Morin, 3,5,7,2′,4′-pentahydroxyflavone, is a yellow polyphenol and abundant in plants of mulberry (*Morus alba* L.), almond (*Prunus dulcis* Mill.) and sweet chestnut (*Castanea sativa* Mill.) [4]. Morin displayed a wide range of therapeutic effects such as UV protection, acting as a strong antioxidant, anti-inflammatory and anti-tumor agent [5]. Morin also shows the ability to reduce colonization of *Staphylococcus aureus* and to inhibit influenza virus in a rat model [6,7]. However, the nature of low aqueous solubility of morin (28.7 mg/L, 37 °C, pH 7.0) means poor bioavailability that tremendously restricts its application in pharmaceuticals and cosmetics [8].

Among various strategies introduced to overcome the limitations of poor soluble drugs and simultaneously enhance dermal absorption for topical application, liposomal vesicles stand out from others because of their biocompatibility and safety profile [9]. Liposomes are spherical vesicles consisting of one or more phospholipid bilayers enclosing an aqueous core. As a result, liposomal carriers can entrap morin into their lipid domains. Additionally, liposomes were shown to create a lipid hydrating film occluding and retaining a higher concentration of drug in the epidermis and dermis, limiting transdermal absorption then reducing unexpected skin irritation [10]. In order to optimize skin penetration of the active compound down to target tissue, deformable liposomal vesicles were employed instead of conventional liposomes as dermal drug delivery systems. These deformable liposomes are also known as transferosomes. They are mainly composed of phospholipids and surfactants. In this study, hydrogenated lecithin and tween-80 were used. Tween-80 acts as an edge activator which destabilizes lipid shells, then increases the elasticity of the vesicles. This not only improves the skin deposition of the drug achieved by the property of traditional liposomes but also promotes deeper skin penetration [11].

Recently, air pollution has been emerging as a global major concern since more than 99% of the entire population are living in areas where levels of pollutants do not meet the standard for healthy air (WHO, 2017). Air pollutants consist of gaseous molecules and solid particles of different chemical compositions, sizes, and shapes, also known as particulate matters [12]. Previous studies have demonstrated that PMs could target various organs in the human body [13]. Skin is the largest organ of the body and acts as a shield defending internal organs against the environmental pollutants. PMs were proved to penetrate into the epidermis to damage the skin barrier function, activate an inflammatory cascade by generating reactive oxygen species (ROS) and oxidative stress [14,15]. Excessive ROS generation would induce phosphorylation of mitogen-activated protein kinase (MAPK) proteins, including extracellular signal-regulated kinase (ERK) and p38, which then results in transcriptional upregulation of matrix metalloproteinases (MMPs) such as MMP-1, leading to a degrading of the extracellular matrix (ECM) [16,17]. Collagen fibers, accounting for the largest portion in structural ECM, become more rigid and fragmented, mainly regulated by MMP proteins during the skin aging process [18]. However, little understanding of morin entrapped in deformable novel liposomal carriers for topical anti-aging benefit has ever been reported.

The present study aimed to prepare deformable forms of liposomal morin with various ratios of compositions, measure their physicochemical features including aqueous solubility, intermolecular bonding, particle size and morphology, and skin absorption; and further investigate the in vitro inhibitory capacity of morin and its liposomal formulation on ROS generation, the underlying mechanism of skin anti-aging property on HaCaT keratinocytes induced by PM.

## 2. Materials and Methods

### 2.1. Materials

Morin hydrate with 90% of purity was purchased from Tokyo Chemical Industry Co., Ltd. (Tokyo, Japan). Lecinolws-50 (lysolecithin; Cosphaderm^®^ E145V)) was purchased from Cosphatec GmbH (Hamburg, Germany). Pluronic F-68 non-ionic surfactant was ordered from Sigma (St Louis, MO, USA). Tween-80 (edge activator) was purchased from Shimakyu’s Pure Chemicals (Osaka, Japan). HaCaT keratinocyte cell line was acquired from the I.Z.S.L.E.R. (Institute Zooprofilattico Sperimentale della Lombardia e dell’Emilia Romagna, Brescia, Italy). Primary antibodies including anti-phospho-ERK 1/2 (Thr202/Tyr204, Thr185/Tyr187) and anti-phospho-p38 (Thr180/Tyr182) were obtained from EMD Millipore Corporation (Temecula, CA, USA). GAPDH antibodies were obtained from Santa Cruz Biotechnology (Dallas, TX, USA) and MMP-1 antibodies were purchased from Proteintech Group, (Rosemont, IL, USA). Fetal Bovine Serum (FBS) and Gibco Dulbecco’s Modified Eagle’s Medium (DMEM) for cell culture were supplied by Thermo Fisher Scientific (Waltham, MA, USA).

### 2.2. Preparation of Liposomal Morin

General methods for preparing liposomal vesicles technically includes four stages named lipid drying, aqueous dispersion, purifying, and analysis [19]. In this study, deformable liposomal morin was developed using the thin film hydration method which obviously involved the processes mentioned above. Firstly, 20 mg of morin was dissolved in 10 mL ethanol and different amounts of lecinolws-50 depending on the ratios in formulations were added to the solutions (Table 1). Tween-80 was then added together under constant mixing by a stirrer bar until a uniform solution was set, which produced phase A. PF-68, 20 mL of distilled water and an appropriate amount of lecithin were mixed using stirrers until completely dissolved in a separate container, which produced phase B. Next, phase B was slowly dropped into phase A by a plastic pipette under homogenizing at 22,000 rpm in a dispersing stage. After 5 min, solvents were removed by a rotary vacuum evaporator and we obtained a dry liposomal film. After that, 10 mL of sterile phosphate buffered saline (PBS) was added with vigorous shaking to rehydrate and reconstitute lamellar vesicles. The mixture then underwent a freezing-drying process at −80 °C for 72 h and further vacuum-drying to evaporate any residual solvent. Liposomal morin was finally obtained and stored at 4 °C for the following experiments.

### 2.3. High Performance Liquid Chromatography (HPLC) Analysis of Morin

Identification and quantification analysis of morin in this research were acquired using HPLC. The HPLC instrument (LaChrom Elite L-2000, Hitachi, Tokyo, Japan) is equipped with an L-2130 pump, an L-2200 autosampler and an L-2420 (UV–vis) detector connecting with a Mightysil RP-18 GP column (250 × 4.6 mm, 5 µm; Kanto Chemical Co., Inc., Tokyo, Japan). The mobile phase consists of acetonitrile and monopotassium phosphate (35:65, *v*/*v*). The system was set at 1 mL/min for flow rate, detecting wavelength was at 355 nm, injection volume was 10 µL and 8 min of running time. Morin was dissolved in ethanol and diluted to a series of nine concentrations ranging from 0.01 to 100 µg/mL in order to determine the calibration curve. Absorption peak of morin was observed at 6.1 min. The calibration curve of morin displayed an excellent linear with *R*^2^ = 0.9999.

### 2.4. Determining the Yield Rate of Liposomal Morin Preparation

The production yield of a drug is worth significant consideration in terms of mass manufacturing. A low yield rate of pharmaceutical production would increase costs for both suppliers and customers. Therefore, it is essential to primarily evaluate the recovery percentage of liposomal morin prepared by the thin film hydration method. Technically, a quantity of each liposomal formulation corresponding to containing 1 mg of morin in theory was totally dissolved in ethanol using a vortex shaker and HPLC was employed to measure concentration of morin. This concentration was then compared to 1 mg of morin for anticipating the recovery percentage after the preparation process.

### 2.5. Determination of Morin and Liposomal Morin Aqueous Solubility

Measuring the solubility of morin and its liposomal formulation is the primary step towards preparing an appropriate concentration for further experiments. Specifically, 1 mg of free morin and liposomal morin that contains equal amounts of morin were dissolved in water (at room temperature) under a vigorous vortex (Gene 2, Scientific Industries, Bohemia, NY, USA) for uniform mixing. Solutions were subsequently filtered through a 0.45 µm syringe filter (13 mm Acrodisc^®^ syringe filters with GHP membrane, Pall Corporation, Port Washington, NY, USA) prior to being ten-fold diluted for HPLC analysis. The above-mentioned standard curve was employed to determine the concentrations of samples and then calculate the aqueous solubility of morin in free form and in liposomal formulations.

### 2.6. Morphology of Liposomal Morin Photographed by Transmission Electron Microscopy (TEM)

TEM is the most frequently used imaging technique for characterizing morphology of nano-scaled particles such as liposomes. In this study, we prepared samples by the negative-staining method due to its simplicity and availability. Each sample of liposomal morin formulations was completely dissolved in de-ionized distilled water to concentration 100 µg/mL. Then 200 µL of prepared solution was slowly dripped onto a 200-mesh copper grid and left to dry. At the same time, phosphotungstic acid (Sigma, St Louis, MO, USA) 0.5% (*w*/*v*) in distilled water was made ready to drop onto dried vesicles (200 µL was used for each copper grid). The resulting samples were stored in the empty plastic capsules until being placed in a transmission electron microscope (JEM-2000EXII instrument; JEOL Co., Tokyo, Japan) for their morphological observation.

### 2.7. Particle Size and Polydispersity Index (P.I.) of Liposomal Morin by Dynamic Light Scattering (DLS)

The sample preparation for particle size measurement included dissolving each formulation in distilled water and diluting sample solutions to an appropriate concentration so that an analyzer could detect the dimension of vesicles. In this research, particle size and size distribution were determined by ELSZ-2000 (Otsuka Electronics, Osaka, Japan).

### 2.8. DPPH^•^ Free Radical Scavenging Assay

DPPH^•^ (2,2-diphenyl-1-picrylhydrazyl hydrate), a stable free radical at room temperature, is a simple screening test to investigate the antioxidant property of morin and liposomal morin in this study. First of all, morin was prepared in de-ionized distilled water and ethanol of an ascending concentration series (10, 25, 50, 100 and 200 µg/mL). Liposomal morin was dissolved in water at the same concentrations. A solution of DPPH^•^ 200 µM was made ready to use by dissolving DPPH^•^ powder in ethanol 95%. Then 100 µL of each sample would react with 100 µL purple DPPH^•^ solution in a 96-well plate. It was noted that the plate was incubated in darkness at room temperature. After 30 min, the optical density of each well at wavelength 517 nm (OD_517_) was measured by a microplate spectrophotometer (SpectraMax ABS^®^ ABS Plus, Molecular Devices, San Jose, CA 95134, USA). Data were collected and processed using the following formula to calculate the free radical scavenging ability of each sample.
$${\mathrm{DPPH}}^{•}\;\mathrm{scavenging}\;\mathrm{activity}\;(\%)=\frac{\mathrm{OD}517\;\mathrm{of}\;\mathrm{control}\;-\;\mathrm{OD}517\;\mathrm{of}\;\mathrm{samples}}{\mathrm{OD}517\;\mathrm{of}\;\mathrm{control}}\;\times \;100\%$$

### 2.9. Intermolecular Bonding of Morin and Liposomes by Fourier Transform Infrared Spectroscopy (FTIR)

Molecular vibration created by radiation absorbed at a specific frequency is used to identify occurring functional groups and chemical interaction of compounds. In this study, FTIR was investigated to figure fundamental chemical groups in the structure of morin, liposomes and their interaction in complexes. Samples including morin, blank liposomes and liposomal morin were handled using the compressed pellet method. Each sample was firstly blended with KBr (Sigma, St Louis, MO, USA) (ratio 1:9 in volume). The mixture was then ground into fine powder using an agate pestle and mortar. Ground powder was transferred to a dry pellet die, pressed into a thin translucent film and carefully placed in a spectrometer (Perkin-Elmer 200 spectrophotometer, Perkin-Elmer, Norwalk, CT, USA). Wavelength range for scanning was set at 400 to 4000 cm^−1^ which covers nearly all of the chemical group spectrums.

### 2.10. In Vitro Dermal Absorption Test

According to the standard guideline of the European Cosmetic Toiletry and Perfumery Association (COLIPA), the Franz Diffusion Cell (FDC) has been employed to measure kinetics of percutaneous absorption of morin and liposomal morin. The FDC system basically consists of two primary chambers: upper donor and lower receptor chamber made with glass, separated by a membrane; in this case pork skin was used. Pork skin samples were purchased from the local traditional market. Areas of skin that did not have the intact barrier or were compromised with any wounds which could result in incorrect data were removed. A 4 cm^2^-(2 cm × 2 cm)-sized pork skin was used for each FDC and placed onto the FDC with the stratum corneum facing upwards; the receptor chamber below was filled with a buffer solution containing 0.14 M NaCl, 2 mM K_2_HPO_4_, and 0.4 mM KH_2_PO_4_ (pH 7.4). A small magnetic stirring bar was placed at the bottom of the receptor chamber in order to maintain the constant compound concentration and temperature by the circulation of water at 32 °C from a water bath. The stirrer speed was set at 600 rpm during the experiment. The sample preparation included dissolving morin, liposomal morin in water comparable to 1 mg of morin per 1 mL of distilled water. Then 200 µL of testing samples were loaded onto the donor chamber in direct contact with the outer-most layer of pork skin. After 1, 2, and 4 h, testing compounds and skin samples were collected for analysis. The stratum corneum layer was removed by stripping 15–20 times with 3 M Transpore tapes. The remaining skin samples were heated at 80 °C to facilitate the separating of the viable epidermis and dermis. They were then cut into smaller pieces with a scalpel. The individual skin layer of each testing sample was soaked into methanol and placed in an ultrasonic bath sonicator for 1 h to extract the remaining morin out of the skin. The extracts were filtered with 0.45 µm pore-sized- membrane before HPLC analysis with the method mentioned earlier.

### 2.11. Cell Culture and Cytotoxicity Assay

Cell metabolism and survival rate play a key role in screening appropriate doses of drug that do not lead to toxicity or cell death but result in therapeutic effectiveness. MTT assay is the most frequently used to measure cellular metabolic activity as an indicator of cell viability, proliferation and cytotoxicity. HaCaT keratinocytes were acquired from Istituto Zooprofilattico Sperimentale della Lombardia e dell’Emilia Romagna (Brescia, Italy). Culture medium included Gibco Dulbecco’s Modified Eagle Medium DMEM (Himedia Laboratories, Mumbai, India), 10% fetal bovine serum (Thermo Fisher Scientific, Naucalpan de Juarez, Mexico) and 1% penicillin–streptomycin (Biological Industries, Connecticut, NE, USA). Cells were grown in an incubator (Thermo Fisher Scientific, Waltham, MA, USA), temperature was set at 37 °C and 5% CO_2_. To begin with, HaCaT cells were seeded in a 96-well plate, specifically, 1 × 10^4^ cells suspended in 100 µL of culture medium per well in 24 h prior to treating. In terms of sample preparation, morin was originally dissolved in dimethyl sulfoxide (DMSO), liposomal morin was prepared in PBS, the solutions were then diluted in DMEM to desired concentrations (10, 25, 40, 100 and 200 µM, respectively). The cells in each well were washed with 100 µL of PBS before being exposed to morin and liposomes for the following 24 h. On the third day, the medium with the active compounds was removed from cells by washing with PBS and 150 µL of 0.5% methyl thiazolyl tetrazolium (MTT) solution was then added into each well. The plate was stored back into the incubator and left for reaction. After 3 h, MTT solution was discharged from wells. Purple formazan crystals created by the reduction of a yellow tetrazolium salt was dissolved in DMSO solution. Optical density at a wavelength of 550 nm (OD_550_) was determined using a microplate spectrophotometer (SpectraMax ABS^®^ ABS Plus, Molecular Devices, San Jose, CA, USA).

### 2.12. PM-Induced ROS Reduction Assay

ROS inhibition is one of the most potential bioeffects of morin and its liposomes in terms of topical application. In order to screen for their impact on the ROS generation, 1 × 10^4^ HaCaT keratinocytes were initially seeded on the 96-well plate and cultured for 24 h with medium solution and incubating conditions mentioned above in the cell culture section. Preparing samples involved dissolving morin in DMSO and liposomal morin in PBS, then diluting to concentrations of 10 and 25 μg/mL by DMEM. Cells were washed with PBS before being treated by the sample preparations for the following 24 h. On the third day of the assay, HaCaT cells were reacted with 20 µM solution of dichlorodihydrofluorescein diacetate (DCFH-DA; Sigma Aldrich, Darmstadt, Germany) for 30 min under an incubation setting. PMs (Standard Reference Material, SRM^®^ 1649b) was purchased from the National Institute of Standards and Technology and suspended in PBS to the concentration of 10 mg/mL. The suspension was sonicated with ultrasound water bath for 10 min. After that, PMs was added to each well (50 µg/cm^2^) and kept exposed for an hour. The cell plate was then washed with PBS and measured for fluorescence absorbance at the excitation and emission wavelength of 485 and 528 nm, respectively, using a microplate reader (BioTek, Winooski, VT, USA). The experiment was triplicated and data were presented by mean value ± SD.

### 2.13. Western Blotting

The anti-aging effect of morin and liposomal morin in keratinocytes were analyzed by Western blotting. Firstly, 4 × 10^5^ HaCaT keratinocytes were seeded in a 6 cm dish and cultured for 24 h in the incubator set with the condition described above. PM were suspended in PBS (10 mg/mL) and sonicated for 10 min. The cells were then treated with morin or liposomal morin prepared in DMEM solution at various concentrations for 3 h. Subsequently, the HaCaT cells were induced with PM suspension (50 µg/cm^2^). After 2 or 6 h, the whole proteins in treated cells were extracted using RIPA Lysis Buffer (Merck Millipore, Burlington, MA, USA) and collected into 1.5 mL tubes, centrifuged at 12,000 rpm for 10 min. The protein contents were then determined by the BCA Protein Assay Kit (Thermo Fisher Scientific, Waltham, MA, USA). An equal quantity of protein in each sample was loaded into sodium dodecyl sulfate-polyacrylamide gel electrophoresis (SDS-PAGE) for separating. Proteins were blotted onto polyvinylidene difluoride (PVDF) membranes (Merck Millipore, Burlington, MA, USA). The membranes were then blocked for an hour and washed with tris-buffered saline (TBS) and 1% tween-20 before being incubated overnight at 4 °C with primary antibodies including p-p38 (1:1000; #09-272, Merck Millipore), p-ERK (1:1000; #05-797R, Merck Millipore), MMP-1 (1:1000; 10371-2-AP, Proteintech Group) and GAPDH (1:1000; sc47724, Santa Cruz Biotechnology, Dallas, Germany) as the internal control. The membranes were washed with TBS prior to being incubated with horseradish peroxidase (HRP) conjugated secondary antibodies which were already reacted with chemiluminescence reagents (ECL; Thermo Fisher Scientific) for an hour at room temperature the following day. Protein signaling bands were achieved by Touch Imager (e-BLOT; Shanghai, China) and analyzed using ImageJ application.

### 2.14. Statistical Analysis

Data were shown as mean ± standard deviation (SD). For protein quantification analysis, ImageJ software (GraphPad Software, La Jolla, CA, USA) was employed. SPSS software (Version 20, IBM Corp., Armonk, NY, USA) was used for statistical analysis. Comparison of data between multiple groups was performed using analysis of variance (ANOVA) followed by Tukey’s post-hoc test, *p* < 0.05 was regarded as statistical significance.

## 3. Results

### 3.1. DPPH^•^ Free Radical Scavenging Assay of Morin

Antioxidant activity is an overall ability to scavenge free radicals, the higher power of neutralizing free radicals, the better the antioxidant property of the component. As we can see in Figure 1, the ability of morin in water to scavenge DPPH^•^ radicals did not exceed 40% at various concentrations since morin was water-insoluble. However, when freely dissolved in ethanol, the scavenging activity of morin at 200 µg/mL was twice as much as morin in water, reaching approximately 70%. In particular, the DPPH^•^ removal effect of morin was concentration-dependent.

### 3.2. Cytotoxicity Assay of Morin

A cell viability assay was conducted to determine the non-toxic concentration range in which more than 80% of keratinocytes still alive after treatment (Figure 2). In this experiment, an MTT assay illustrated that almost 100% of HaCaT cells maintained their usual metabolization and proliferation when treated with morin at concentration 10, 25, 40 and 100 µM. In contrast, nearly 30% of cells were dead in the group exposed to 200 µM of morin for 24 h. Therefore, in further experiments, appropriate dosage used to investigate in vitro bioactive effects of morin should be lower than 100 µM for the purpose of cellular non-toxicity.

### 3.3. Morin Inhibited PM-Induced ROS Generation

As shown in Figure 3, PM increased the ROS level in HaCaT cells by nearly two and a half times when compared to the negative control group. Two concentrations of morin used in this assay were 10 µM and 25 µM. The inhibitory effect of morin on ROS generation was better achieved by 25 µM, which reduced the ROS level to approximately a half compared to the PM group.

### 3.4. Morin Downregulated Expression of Proteins in MAPKs

As shown in Figure 4, PM obviously increased the levels of p-ERK and p-p38 in HaCaT keratinocytes by nearly three times. However, in cells first exposed to PM and then treated with morin, their p-ERK and p-p38 expression significantly decreased. 25 and 40 µM of morin were effective in reducing the expression of p-ERK and p-p38 in HaCaT keratinocytes induced by PM.

### 3.5. Morin Suppressed MMP-1 Expression

Regards to anti-aging activity, we examined the effect of morin on the suppression of MMP-1 (Figure 5). HaCaT keratinocytes were first pretreated with morin at 25 and 40 µM for 3 h then induced with PM for 6 h. As we can see in Figure 5, 40 µM of morin successfully reduced the transcription of MMP-1 by nearly two-thirds with a significant difference (*p* < 0.05). The inhibitory effect of 25 µM on MMP-1 was small. For more details about protein expression, the percentage change was calculated and is shown in Table 2.

### 3.6. Aqueous Solubility of Morin and Its Liposomal Formulations

Table 3 displays the solubility of morin and its liposomal formulations in water with various ratios. Looking at the table, there was very little morin dissolved in water—less than 2 mg/L (<0.1 mg/mL), demonstrating that morin was practically aqueous insoluble, according to the United States Pharmacopeia (USP). In contrast, morin-carried liposomes showed hundred-time higher water solubility than morin in free form. The highest value could reach up to approximately 475 mg/L with formulation 6 (1:10:20:5) and the lowest was nearly 290 mg/L with formulation 8 (1:2.5:2.5:10). Generally, those formulations with a larger amount of lecithin such as formulation 1 (1:20:20:5), formulation 2 (1:40:20:5) and formulation 3 (1:80:20:5) were less aqueous soluble than ones with lower phospholipid content including formulations 4, 5 and 6. Moreover, when comparing between formulations 7 and 8, the addition of PF-68 did not improve the water solubility of the vesicles.

### 3.7. Yield Rate of Liposomal Morin Preparation

In contrast to the water solubility, the yield rate showed an obvious proportional dependence on the lecithin portion in lipid bilayers (Table 4). The total amount of liposomal morin ranged from 57% to nearly 80% with formulations 8 and 3, respectively. This could be explained by the viscosity of phospholipid in the formulations which could help retain the active ingredient in the final products. Consequently, the thin film hydration method produced a moderate to high yield and a higher recovery percentage would be obtained by involving a larger quantity of lecithin into the formulation.

### 3.8. Morphology of Liposomal Morin Photographed by TEM

Morphological imaging of LM was taken by TEM (Figure 6). As we can see in those photos with magnification of ×30.0k, liposomal vesicles were found in all formulations but they varied in their size and bilayer shape. Formulations 2 and 3, containing the highest proportion of phospholipid, seemed to have the most well-defined spherical vesicles. However, the liposomes formed larger aggregations in formulation 3 and this phenomenon was also seen in formulation 6. Besides, particle size was bigger with formulation 2 and 5, approximately 600–650 nm. Smaller and deformed liposomes were found in formulation 1 and 7, ranging from 50 to 250 nm, vesicles were less aggregating.

### 3.9. Particle Size and Polydispersity Index (P.I.) of Liposomal Morin by DLS

Table 5 shows the diameter of LM formulations reconstituted in water. It can be seen that there was an increase in particle size with samples containing a higher level of lecinolws-50, tween-80 and PF-68, ranging from 383.8 ± 31.1 nm to 597.3 ± 63.0 nm for formulations 3 and 2, respectively. Vesicles prepared with less phospholipid and other surfactants were significantly smaller in diameter, varying from 229.5 ± 35.9 nm to 276.1 ± 16.9 nm for formulations 1 and 8. Besides, size distribution is an important characteristic of liposomal vesicles and could be examined by P.I. A high value of P.I. close to 1 indicates a broad variation in size or presence of vesicular aggregation. It can be seen that formulation 3 produced the most uniform liposomes due to the smallest P.I. (0.270 ± 0.026). In contrast, a wide range of particle size with P.I. 0.656 ± 0.056 was in formulation 8. Other formulations including 1, 2 and 7 resulted in acceptable values of P.I. for lipid-based carriers, which were 0.354, 0.355 and 0.389, respectively.

### 3.10. Intermolecular Bonding of Morin and Liposomes by FTIR

FTIR spectra of morin, liposomal morin formulations 1 and 7, blank liposome formulations 1 and 7 are illustrated in Figure 7. Broad and intense adsorption peak at 3449 cm^−1^ represents dimeric hydroxyl O–H stretching functional. A pair of absorptions at 1620 cm^−1^ is indicative of double bond alkenyl C=C stretching. Peaks between 1600 and 1500 cm^−1^ signify one or more aromatic ring vibrations, specifically, 1511 cm^−1^ was assigned to C=C–C ring bonding. Band at 1177 cm^−1^ suggests a phenolic C–O stretching vibration. A peak at 831 cm^−1^ was a C–H vibration of aromatic ring. A small sharp peak at 639 cm^−1^ was attributed to a bending vibration of the OH alcoholic group. P–O bending vibration peaks were demonstrated at 539, 538, 586, 558 cm^−1^ in liposomal morin and empty liposome preparation, which contain lecithin as a component and not seen in the infrared spectrum of morin. C–H stretch vibrations for methyl (–CH_3_) or methylene (–CH_2_–) peaking at 2922, 2923, 2925, and 2927 cm^−1^ are distinctive in saturated or unsaturated aliphatic compounds that were depicted in all graphs, except for morin. As we can see that the infrared spectrum of blank liposome formulations 1, 7 (BLF1, BLF7) and morin were merged to shape the spectrum of liposomal morin formulation 1, 7 (LMF1, LMF7), respectively. Even though some peaks shifted to higher or lower wavelengths, no noticeable new peaks were found in the spectra of LMF1, LMF2. Therefore, morin was supposed to be successfully encapsulated in liposomes without unexpected chemical reactions between ingredients.

### 3.11. In Vitro Dermal Absorption Test

Figure 8 illustrates the penetration of LMF1, LMF7 through the stratum corneum, viable epidermis (epidermis without stratum corneum) and dermis of pork skin using the FDCs. The total dermal absorption is considered as the amount of substance found in the viable epidermis and dermis. Morin dissolved in water technically could not overcome the skin barrier to reach deeper layers. There was no morin found in the skin samples and therefore not shown on the diagram. In contrast, LM greatly improved the skin absorption of the active and were mainly accumulative in epidermis. Morin concentrations in corneocyte layers were determined at relatively 4–6 µg/cm^2^ for LMF7 and LMF1, respectively. Most of the liposomal morin could overcome the stratum corneum to be deposited in viable epidermis, nearly 50 µg/cm^2^ of LMF1 and 60 µg/cm^2^ of LMF7 were quantified after 4 h of application. A dynamic and time-dependent penetration was most obviously seen in the dermis, approximately 3, 7 and 25 µg/cm^2^ of LMF7 were penetrated into this layer after 1, 2 and 4 h, respectively. As a result, encapsulating morin into liposomes evidently enhanced dermal absorption compared to free morin. No content of morin found in the receptor fluid (data not shown) illustrated liposomes hardly penetrated deeper than the dermal layer into systemic circulation which might cause undesirable adverse effects. LMF7 was superior to LMF1 in total dermal absorption and would be used for further bioactive investigation.

### 3.12. DPPH^•^ Free Radical Scavenging Assay of Liposomal Morin

Figure 9 indicates that encapsulating morin into liposomes showed a great benefit to increase water solubility of morin and therefore significantly enhanced its effect on scavenging DPPH^•^ radicals, ranging from 50 to 60%. However, there was no improvement in the effectiveness of removing DPPH^•^ radicals when increasing LMF7 dosage. This was different from the results of the above experiment in which morin worked in a dose-dependent manner.

### 3.13. Cytotoxicity Assay of Liposomal Morin

Figure 10 demonstrates the viability of HaCaT cells treated with LMF7. Proliferation of keratinocytes was elevated when treated with LMF7 at concentrations 10, 25 and 40 µM. However, significant cell death was seen in the group treated with 100 µM of LMF7. There was only a 26% survival rate in this group. As a result, two concentrations of 25 and 40 µM would be used in further assays including ROS inhibition and skin anti-aging.

### 3.14. Liposomal Morin Inhibited PM-Induced ROS Generation

LMF7 at 25 µM showed a significant benefit to decrease oxygen-containing free radicals by nearly a half compared to the PM-induced group without treating with LMF7 (Figure 11). Similarity to the free morin dissolved in DMSO, at 10 µM, LMF7 did not show any reducing effect on ROS level. Therefore, the baseline concentration of LMF7 for further Western blotting assay would be 25 µM.

### 3.15. Liposomal Morin Downregulated Expression of Proteins in MAPKs

Figure 12 illustrates the inhibitory effect of LMF7 on the expression of proteins in the MAPK pathway including p-ERK and p-p38, respectively. The activation of proteins was considerably higher in the HaCaT cells group that were only treated with PM. In contrast, the expression of p-ERK and p-p38 was obviously reduced in cells that were first induced by PM and then administered 25 and 40 µM of LMF7. However, it could not be concluded if the action of LMF7 was dose-dependent or not.

### 3.16. Liposomal Morin Suppressed MMP-1 Expression

Figure 13 shows that LMF7 displayed a considerable suppression of MMP-1 level in HaCaT keratinocytes. 40 µM of LMF7 greatly inhibited transcription of MMP-1 by approximately three times compared to the PM group while 25 µM inhibited the expression of MMP-1 by nearly one and a half times.

For more details about protein expression, percentage change was calculated and is shown in Table 6.

## 4. Discussion

Airborne PM exposure has become a noticeable risk factor related to many cutaneous disorders in the short- or long-term such as skin allergies, acne, urticaria, eczema, contact dermatitis, dermographism, seborrhea, infected skin disease, psoriasis, dyspigmentation, skin cancer and aging [20]. Dijkhoff et al. elucidated different underlying mechanisms of fine particles impacting skin through exogenous ROS induction. An imbalance of antioxidants and ROS overproduction was thought to activate MMPs via upregulating phosphorylation of proteins in the MAPK signaling pathway, which eventually promoted the degradation of collagen fibers and worsened skin aging [14]. From a cosmeceutical perspective, incorporating plant antioxidants into skin care formulations for anti-pollution and anti-aging effects has drawn a lot of attention recently. Morin, a polyphenol, displays many biological activities that can be applied to treat various skin problems. Yong and Ahn demonstrated that sunscreen containing morin successfully diminished cytotoxicity, prevented damage of UVA radiation on in vitro fibroblasts in a concentration range of 20 and 50 μM [21]. However, no evaluation of morin on the ROS inhibition and anti-aging ability in keratinocytes exposed to PM has yet been carried out.

In this study, we first screened the antioxidant property of free morin dissolved in DMSO by in vitro DPPH^•^ assay. Morin was found to be a powerful antioxidant that could strongly scavenge DPPH^•^ radicals. We further looked at its ROS inhibitory effect in HaCaT keratinocytes exposed to PM and clarified the underlying mechanism of its action on skin antiaging. Our study proved that morin dissolved in DMSO successfully reduced the ROS level and significantly diminished the expression of MAPK proteins including p-ERK and p-p38 in HaCaT cells. Downstream signaling of p-ERK and p-p38 subsequently resulted in suppressing MMP-1 expression. MMPs play the most critical role in ECM degradation throughout the skin aging process. MMP-1, which is mainly secreted by keratinocytes, is especially responsible for breakdown and fragmentation of skin collagen fibers [22]. In previous studies, morin was also proved to decrease phosphorylation of p53, JNK and p38 in keratinocyte stem cells radiated by UVB light [23]. With the above-mentioned findings in present study, we strongly believe that morin is a bright candidate for topical anti-aging and anti-pollution skin care.

However, low bioavailability resulted from poor aqueous solubility of morin is the biggest obstacle for its application and this is also the problematic issue for most of the other flavonoids [24]. Active compounds usually have to be dissolved in solution for better bioavailability and dermal absorption. Water is the most frequently used solvent in pharmaceutical preparations. In addition, water-insoluble drugs often require very high doses in order to achieve desired concentrations for therapeutic effects which results in rising cost, time and burden for manufacturers and consumers [25]. Improving the aqueous solubility of morin is therefore of paramount importance prior to formulating it into topical products. In this study, we overcame the barrier by encapsulating morin into lipid domains of liposomes. Since morin is an amphipathic molecule with presence of aromatic rings and hydroxyl groups, it is entrapped within bilayer shells or between surface of hydrophobic and polar parts of the vesicles [26]. We used the thin film hydration method based on the conventional Bangham method to prepare liposomal morin [27]. This is a simple and cost-effective approach. The whole process involved several straightforward steps such as the dissolution of lipid, solutes and surfactants; homogenization; drying and hydration. Nevertheless, we achieved favorable yield values ranging from nearly 60 to 80%. Production loss was limited with an increasing proportion of lecithin in the formulations. A similar outcome was also found in a previous study [28]. This is especially worth considering when it comes to cost effectiveness in mass production.

Entrapping morin into lipid carriers resulted in water solubility enhancement by nearly 300 to 500 times in this study. Liposomal formulations of other flavonoids including resveratrol, fisetin, quercetin, curcumin and epigallocatechin gallate (EGCG) were also reported to have significantly better water solubility [29,30]. Liposomal flavonoids limited the use of organic solvents which not only presents a better safety profile but also extends the tolerated dose for their optimal therapeutic effects. Furthermore, liposomal vesicles of flavonoids were proved to stabilize the compounds then prolong their efficacy [31].

Previous studies pointed out that dermal absorption of liposomes was inversely related to their particle size. They illustrated that liposomes with a size over 600 nm were unable to penetrate through the stratum corneum and vesicles that were smaller than 70 nm obtained the best deposition in the epidermal and dermal layer. The author also mentioned the significant role of pilosebaceous units in the skin penetration of liposomal vesicles [32]. As a result, formulations with smaller particles were preferable for topical application and further experiments. In our experiments, formulations 1 (1:20:20:5) and 7 (1:2.5:2.5:5) produced the smallest liposomes examined by both DLS and TEM. Moreover, these liposomes were more deformable in shape which was probably due to the incorporation of a similar ratio of phospholipid and tween-80 as an edge activator. These formulations were then applied onto pork skin to investigate their skin penetration.

Our data showed that liposomal morin greatly penetrated deep into the dermis while the free form of morin could not overcome the skin barrier—stratum corneum. In addition, the total dermal absorption of liposomal morin was shown to be not only size-dependent but also highly improved when the lipid outer layers of vesicles were made more flexible with appropriate ratio of lecithin and tween-80. Even though particles from formulation 7 were bigger than from formulation 1 in diameter, lower amounts of lecithin in formulation 7 could produce liposomes with more elastic and deformable lipid shells. This characteristic was thought to help LMF7 penetrate better through stratum corneum and hair follicles leading to greater dermal absorption. Moreover, a majority of liposomal vesicles deposited in epidermis and dermis without travelling to the receptor fluid which emphasized the safety property of liposomes not reaching the systemic circulation.

As a result, formulation 7 would be more preferable to treating HaCaT keratinocytes in further experiments. Cytotoxicity of LMF7 was evaluated beforehand. Although the proliferation of HaCaT keratinocytes was high if treated with 100 µM of morin, a significant reduction in cell growth was seen when treated with LMF7 with concentrations over 50 μM (data not shown). This cell death could be explained by the presence of large amounts of lecithin and tween-80 in the formulations. Vater, et al. also reported that a nanoemulsion containing 5% of tween-80 led to a complete cell death of fibroblasts after 24 h exposure [33]. Therefore, two concentrations, 25 μM and 40 μM of morin and LMF7, were used to evaluate inhibitory capacity on PM-induced ROS, expression of p-ERK, p-p38 and MMP-1 in HaCaT cells. Our study indicated that LMF7 displayed a more powerful suppression on MMP-1 compared to morin in free form, demonstrating that liposomal vesicles could act as the carriers and that successfully released morin to provide skin anti-aging activity.

## 5. Conclusions

To summarize, in this study, we prepared eight formulations of liposomal morin with different ratios of morin, lecinolws-50, tween-80 and PF-68. A significant improvement in water solubility of morin loaded in liposomes was achieved. Deformable particles with a small diameter contributed to considerable skin penetration enhancement. Both morin and liposomal morin effectively protected keratinocytes against ROS generation induced by PM exposure and subsequently provided skin anti-aging and anti-pollution property by reducing MMP-1 expression via downregulating the MAPK signaling pathway. As a result, we suggest that liposomal morin can be applied in various topical products for anti-aging and anti-pollution benefits on skin.

## Figures and Tables

**Figure 1 antioxidants-11-01183-f001:**
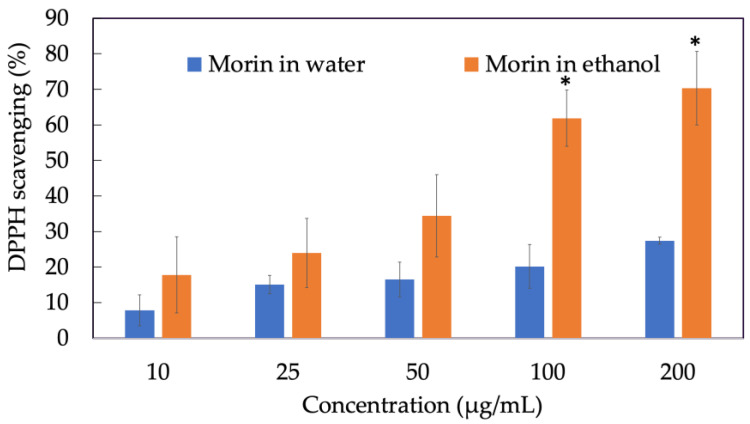
DPPH^•^ radical scavenging activity of morin in water and in ethanol. Data were from triplicate experiments and presented as mean ± SD. * indicates a statistically significant difference between morin in water and morin in ethanol, *p* < 0.05.

**Figure 2 antioxidants-11-01183-f002:**
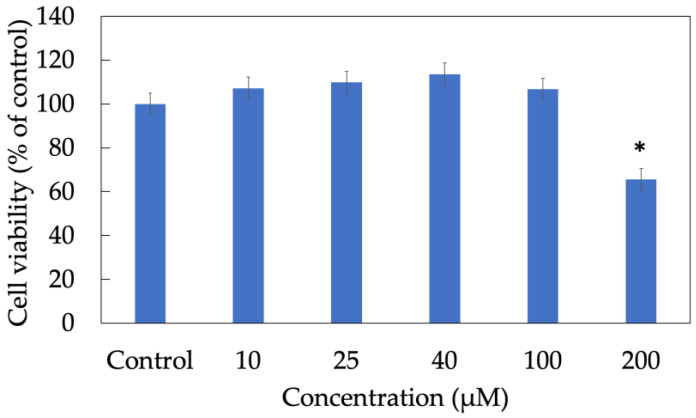
Cell viability of HaCaT keratinocytes treated with morin. Data were from triplicate experiments and presented as mean ± SD. * indicates a statistically significant difference with control, *p* < 0.05.

**Figure 3 antioxidants-11-01183-f003:**
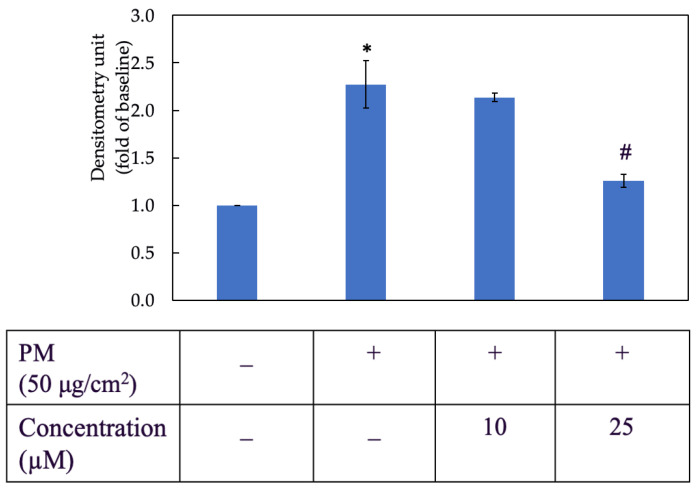
Effect of morin on PM-induced ROS generation in HaCaT keratinocytes. +, − mean with or without exposing to PM or morin, respectively. Data were presented as mean ± SD and from triplicate experiments. *, # indicates a statistically significant difference with control group (without treating PM and morin) and PM group (only treating PM), respectively, *p* < 0.05.

**Figure 4 antioxidants-11-01183-f004:**
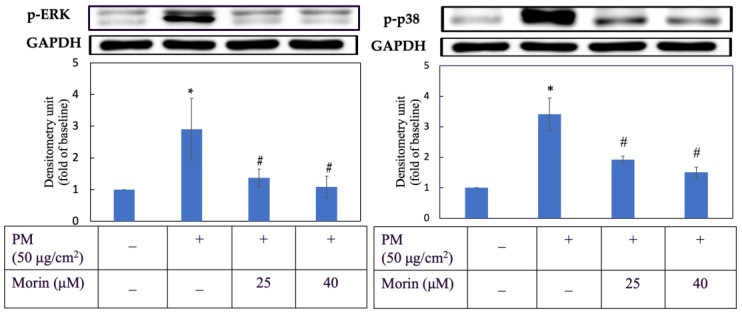
p-ERK and p-p38 expression. Cells were pre-incubated with morin for 3 h and then treated with PM for 2 h. +, − mean with or without exposing to PM or morin, respectively. Data were presented as mean ± SD and from triplicate experiments. *, # indicates a statistically significant difference with control group (without treating PM and morin) and PM group (only treating PM), respectively, *p* < 0.05.

**Figure 5 antioxidants-11-01183-f005:**
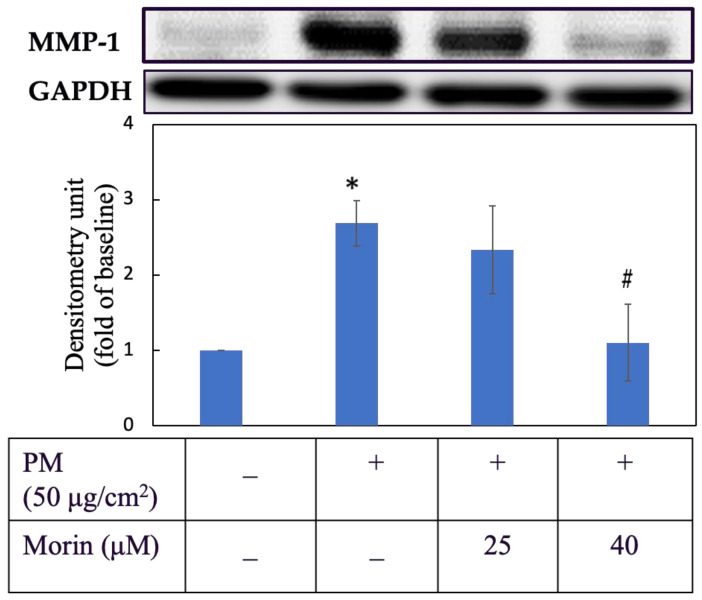
MMP-1 protein expression. Cells were pre-incubated with morin for 3 h and then treated with PM for 6 h. +, − mean with or without exposing to PM or morin, respectively. Data were presented as mean ± SD and from triplicate experiments. *, # indicates a statistically significant difference with control group (without treating PM and morin) and PM group (only treating PM), *p* < 0.05.

**Figure 6 antioxidants-11-01183-f006:**
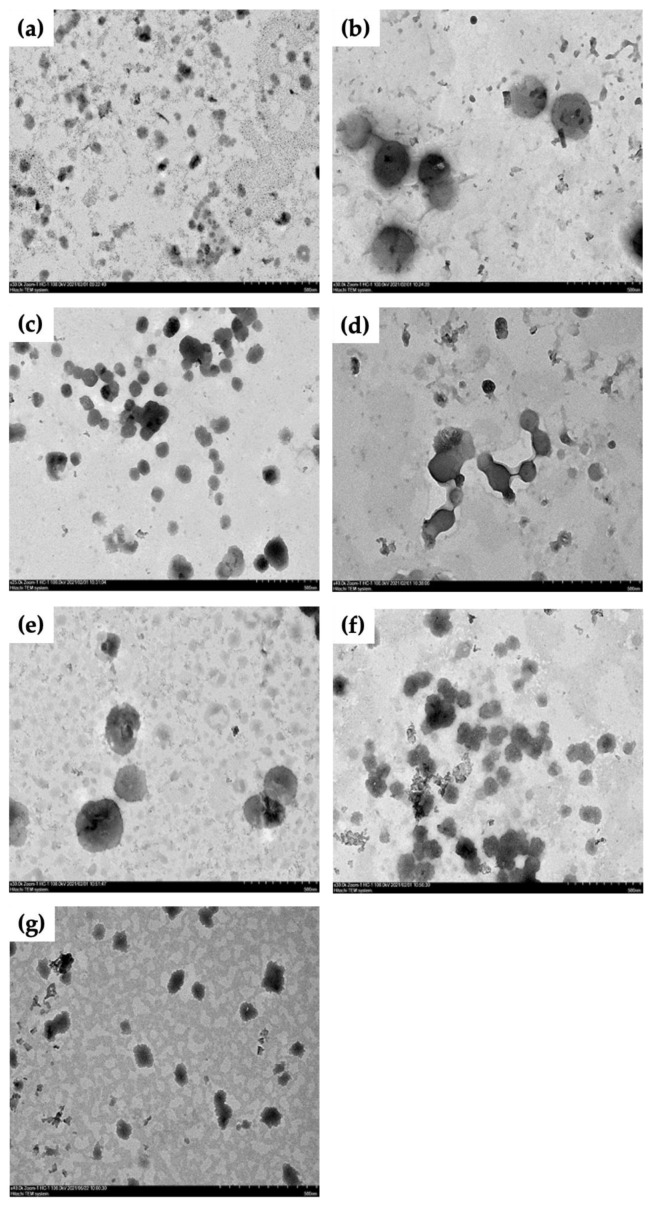
TEM images of liposomal morin with different ratios of morin:lecithin:tween-80, (**a**) (F1) 1:20:20, (**b**) (F2) 1:40:20, (**c**) (F3) 1:80:20, (**d**) (F4) 1:1:20, (**e**) (F5) 1:5:20, (**f**) (F6) 1:10:20, (**g**) (F7) 1:2.5:2.5. Images of samples are magnified by ×30.0k. Scale bar is 500 nm.

**Figure 7 antioxidants-11-01183-f007:**
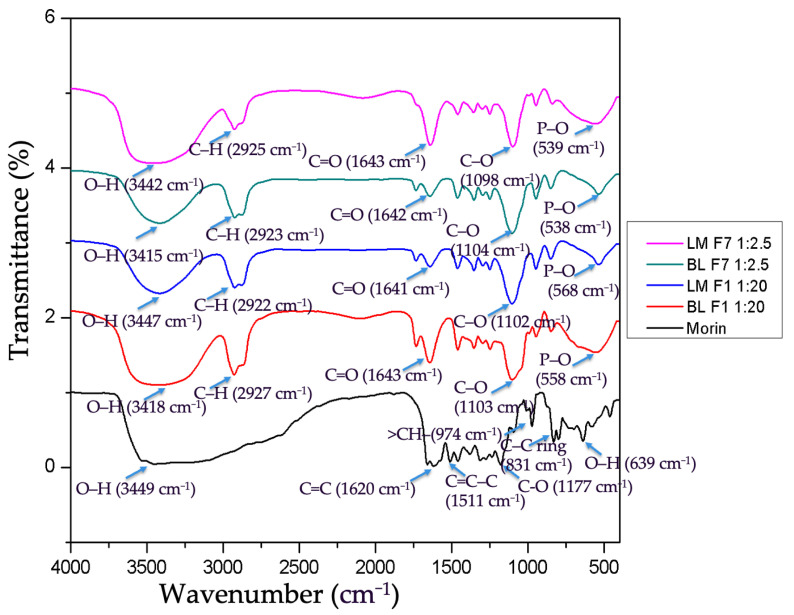
FTIR spectra of morin, liposomal morin formulation 1 (LMF1), 7 (LMF7); blank liposome formulation 1 (BLF1), 7 (BLF7).

**Figure 8 antioxidants-11-01183-f008:**
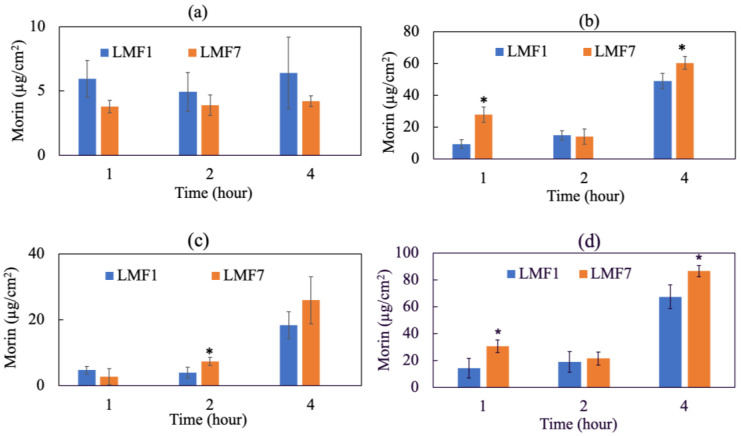
Skin penetration of liposomal morin formulation 1 (1:20:20:5) (LFM1) and 7 (1:2.5:2.5:5) (LMF7) in water using the FDC method and pork skin: (**a**) stratum corneum, (**b**) viable epidermis (epidermis without stratum corneum), (**c**) dermis and (**d**) total dermal absorption. There was no morin found in the receptor fluid. Data were presented as mean ± SD and from triplicate experiments. * indicates a statistically significant difference between LMF1 and LMF7, *p* < 0.05.

**Figure 9 antioxidants-11-01183-f009:**
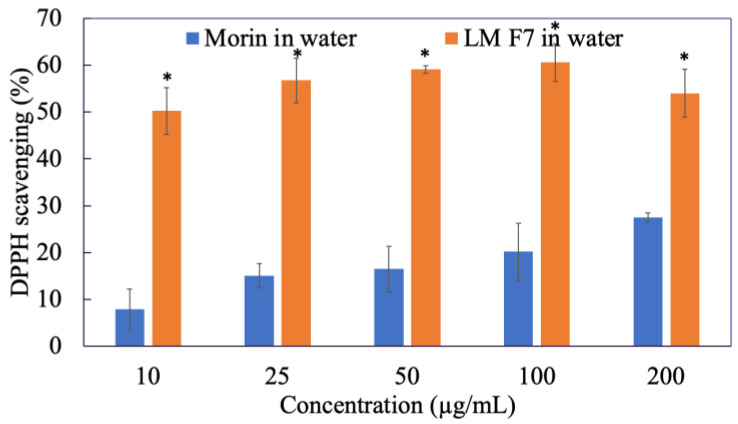
DPPH^•^ radical scavenging activity of morin in water compared to liposomal morin formulation 7 (LMF7). Data were presented as mean ± SD and from triplicate experiments. * indicates a statistically significant difference between morin and LMF7 in water, *p* < 0.05.

**Figure 10 antioxidants-11-01183-f010:**
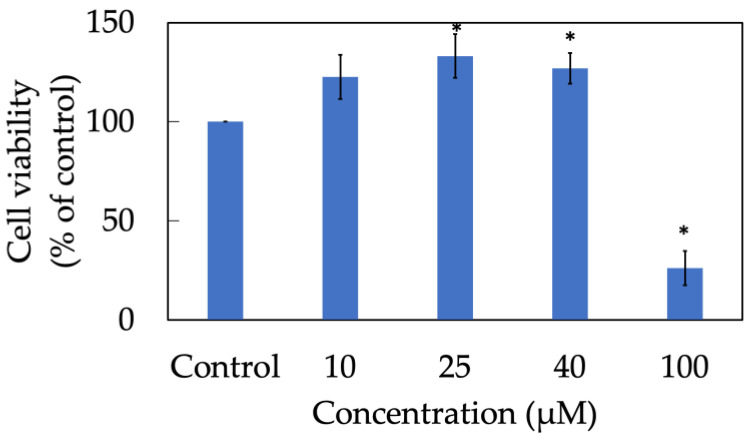
Cell viability of HaCaT keratinocytes treated with liposomal morin formulation 7. Data were presented as mean ± SD and from triplicate experiments. * indicates a statistically significant difference with control group, *p* < 0.05.

**Figure 11 antioxidants-11-01183-f011:**
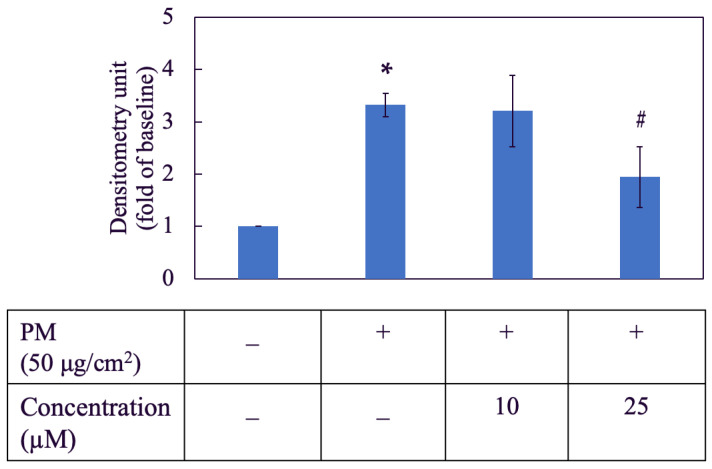
Effect of liposomal morin formulation 7 on PM-induced ROS generation in HaCaT keratinocytes. +, − mean with or without exposing to PM or LMF7. Data were presented as mean ± SD and from triplicate experiments. *, # indicates a statistically significant difference with control group (without treating PM and LMF7) and PM group (only treating LMF7), respectively, *p* < 0.05.

**Figure 12 antioxidants-11-01183-f012:**
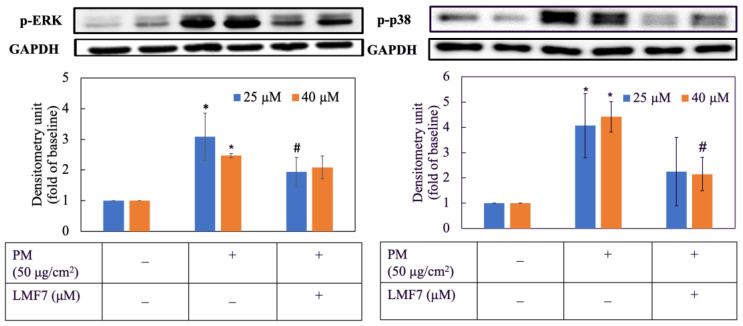
p-ERK and p-p38 protein expression. Cells were pre-incubated with blank liposome or liposomal morin formulation 7 for 3 h and then treated with PM for 2 h. +, − mean with or without exposing to PM or LMF7, respectively. Data were presented as mean ± SD and from triplicate experiments. *, # indicates a statistically significant difference with control group (without treating PM and LMF7) and PM group (only treating LMF7), respectively, *p* < 0.05.

**Figure 13 antioxidants-11-01183-f013:**
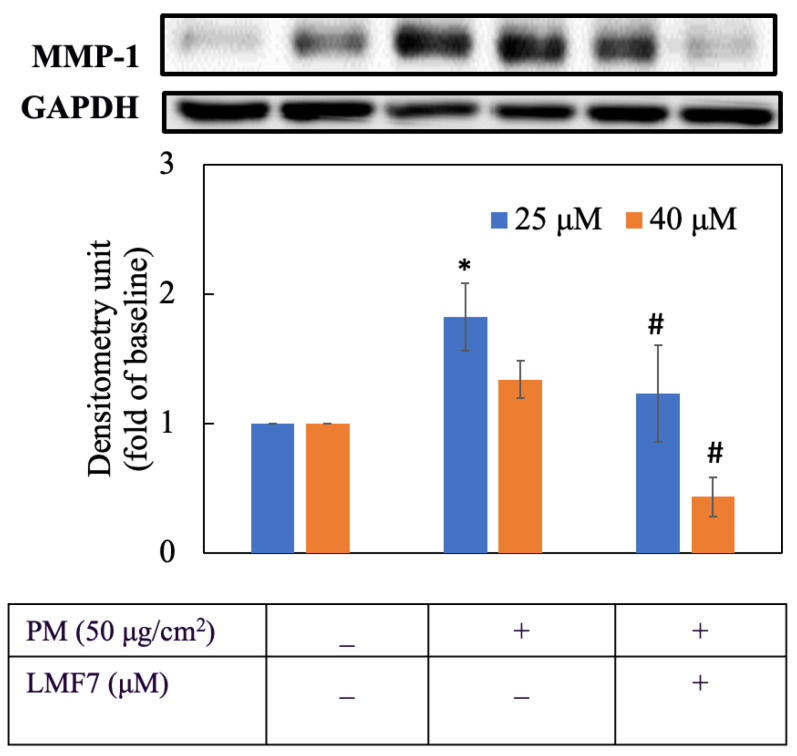
MMP-1 protein expression. Cells were pre-incubated with blank liposome or liposomal morin formulation 7 for 3 h and then treated with PM for 6 h. +, − mean with or without exposing to PM or LMF7, respectively. Data were presented as mean ± SD and from triplicate experiments. *, # indicates a statistically significant difference with control group (without treating PM and LMF7) and PM group (only treating LMF7), respectively, *p* < 0.05.

**Table 1 antioxidants-11-01183-t001:** Ingredients of liposomal morin.

Phase	Ingredients	Formulation							
		1	2	3	4	5	6	7	8
A	Lecinolws-50 (mg)	200	400	800	10	50	100	25	25
	PF-68 (mg)	100							200
	Distilled water (mL)	20							
B	Lecinolws-50 (mg)	200	400	800	10	50	100	25	25
	Morin hydrate (mg)	20							
	Tween-80 (mg)	400						50	50
	Ethanol (mL)	10							
C	PBS (mL)	10							

**Table 2 antioxidants-11-01183-t002:** Percentage change of p-ERK, p-p38 and MMP-1 expression in PM-induced HaCaT cells treated with morin.

Protein	Percentage Change (%) ^a^	
	Concentration of Morin	
	25 µM	40 µM
p-ERK	81.24 ± 8.46	96.31 ± 4.3
p-p38	60.14 ± 10	77.71 ± 9.13
MMP-1	NE ^b^	97.52 ± 25.86

^a^ Percentage change was calculated from quantitative results of Figure 4 and Figure 5 and by following equation: |Fold change^PM^ –Fold change^MD^|/|Fold change^PM^ –Fold change^Con^| × 100%. Data were presented as mean ± SD and from triplicate experiments. ^b^ NE: no effect.

**Table 3 antioxidants-11-01183-t003:** Water solubility of liposomal morin compared to morin.

Formulation	Ratio Morin:Lecinolws-50:Tween-80:PF-68	Water Solubility (mg/L)
Morin		1.7 ± 1.1
1	1:20:20:5	353.9 ± 14.2 *
2	1:40:20:5	353.2 ± 5.4 *
3	1:80:20:5	357.5 ± 30.5 *
4	1:1:20:5	459.0 ± 46.2 *
5	1:5:20:5	467.8 ± 12.5 *
6	1:10:20:5	475.2 ± 1.0 *
7	1:2.5:2.5:5	339.1 ± 53.3 *
8	1:2.5:2.5:10	289.1 ± 21.1 *

Data were presented as mean ± SD and from triplicate experiments. * indicates statistically significant difference with raw morin dissolved in water, *p* < 0.05.

**Table 4 antioxidants-11-01183-t004:** Yield rate of liposomal morin preparation.

Formulation	Ratio Morin:Lecinolws-50:Tween-80:PF-68	Yield Rate (%)
1	1:20:20:5	69.6 ± 1.2
2	1:40:20:5	76.5 ± 0.8
3	1:80:20:5	79.1 ± 1.4
4	1:1:20:5	65.9 ± 0.9
5	1:5:20:5	69.3 ± 7.3
6	1:10:20:5	72.2 ± 1.2
7	1:2.5:2.5:5	62.6 ± 0.3
8	1:2.5:2.5:10	57.0 ± 1.2

Data were presented as mean ± SD and from triplicate experiments.

**Table 5 antioxidants-11-01183-t005:** Diameter and polydispersity index of liposomal morin with different ratios.

Formulation	Ratio Morin:Lecinolws-50:Tween-80:PF-68	Diameter (d) (nm)	Polydispersity Index (P.I.)
1	1:20:20:5	229.5 ± 35.9	0.354 ± 0.055
2	1:40:20:5	597.3 ± 63.0	0.355 ± 0.030
3	1:80:20:5	383.8 ± 31.1	0.270 ± 0.026
7	1:2.5:2.5:5	246.2 ± 12.2	0.389 ± 0.006
8	1:2.5:2.5:10	276.1 ± 16.9	0.656 ± 0.056

Data were presented as mean ± SD and from triplicate experiments.

**Table 6 antioxidants-11-01183-t006:** Percentage change of p-ERK, p-p38 and MMP-1 expression in PM-induced HaCaT cells treated with liposomal morin formulation 7.

Protein	Percentage Change (%) ^a^	
	Concentration of LMF7	
	25 µM	40 µM
p-ERK	53.46 ± 16.81	26.53 ± 17.38
p-p38	68.30 ± 32.42	65.18 ± 16.38
MMP-1	73.93 ± 31.43	313.25 ± 139.91 *

^a^ Percentage change was calculated from the quantitative results of Figure 12 and Figure 13 and by the following equation: |Fold change^PM^ –Fold change^PBS^|/|Fold change^PM^ –Fold change^Con^| × 100%. Data were presented as mean ± SD and from triplicate experiments. * indicates a statistically significant difference with 25 µM group, *p* < 0.05.

## Data Availability

The data presented in this study are available in article.

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
