# Peer review of "Liposomes Encapsulating Morin: Investigation of Physicochemical Properties, Dermal Absorption Improvement and Anti-Aging Activity in PM-Induced Keratinocytes"

_antioxidants, 2022, doi:10.3390/antiox11061183_

Round 1

Reviewer 1 Report

Dear authors,

generally this manuscript is interesting, but has to be modified at some insatnces to be suitable for publciations. 

General remarks:

1) The outline of the mansucript is not 100% clear to me. The most common form for topical apllication onto the skin is the inclusion into creams. Morin would be perfectly suitetd for that, even without liposmes. It is stated in the text, that topical application is mainly based on aqueos solutions in pharamcy...I heavily doubt that. 

2) As correctly indicated in the text: ROS stimulate the activity and expression of MMPs. This leads to degardation of collagen and fianlly wrinkles. Rhe cell type that is mainly affected by that are fibroblasts. Why were keratinocytes used for this study?

3) Statistics at most instances is absent. This has to be corrected. Each figure with qunatitaive data needs statistsics. the same is ture for all tables.

4) The image quality (resolution) at many instances is horrible and the tetx is very often close to be unreadable.

5) Most of the time is good, but has to be improved at many instances. A good example ist the sentecne in line 60 to 64.

6) The pharse "on the other hand" is used very often. the corerct usage would be "on the one hand....on the other hand". Both are linked.

7) Species names are not always italic.

8) Formulas in line 142 and line 250 are basic math and should be removed.

9) I know that some measurements lead to survival rates above 100%. It is somehow okwards that the 110% survival (that is impossible) is especailly mentioned in the text (it is fine for the figure) (line 315).

10) line 318: What doess abnormally grew mean?

11) line 325: What does "induced with PM by increasing ROS" mean?

12) explain + and # in all figures when used. I assume * means normalized to control without PM, # to PM without morin????

13) The expmnation of the WB in the text is WRONG: line 338ff: It is not a downregualtion of ERK and p38 but a decraesed phosphorlyation. Or in other words: the english at that section should be corrected/explained in more detail

14) Table 2: The same as in 13. It ist not only protein expression but also the phosphorylation status. To celarly distinguish two antibodies should have been used (one for ERK, one for p-ERK).

15) The quality of Figure 6 should be improved:  Same distances between images itself and between images and text

16) Eventually the effect of polydiversity on the interperation of the results should be explained more deeply. The abbreavation P.I should be explained. I assume P.I index is wrong (line 410). It should be either P. index or only P.I.

17) There is some discrepancy between the absorption test and the biological assays. The abosrption ets was performed on(/in dermis, whereas all other assay aimed on epidermis/keratinocytes. That should be commented.

Author Response

Dear Reviewer,

 Thank you for your thorough comments and useful suggestion. Below is the author’s response, and we hope that it could clarify your remarks

Best Regards,

Yen FL

 1) The outline of the mansucript is not 100% clear to me. The most common form for topical apllication onto the skin is the inclusion into creams. Morin would be perfectly suitetd for that, even without liposmes. It is stated in the text, that topical application is mainly based on aqueos solutions in pharamcy...I heavily doubt that.

Reply:

  • I totally agree with you that cream is one of the most commonly used topical dosage forms
  • Any drug to be absorbed must be present in the form of solution at the site of absorption
  • Poor water-soluble drugs exhibit low skin penetration and permeation, meaning insufficient bioavailability that can not produce desirable effects.
  • Liposome is not the only but one of various approaches to improve solubility of drug. Otherwise, solvents, surfactants and solubilizers can be used, but they are more likely to cause skin irritation.
  • I made myself clearer in line 47-49, thank you for your suggestion.
  • “However, the nature of low aqueous solubility of morin (28.72 + 0.97 µg/ml, 37 °C, pH 7.0) means poor bioavailability that tremendously restricts its application in pharmaceuticals and cosmetics”.

2) As correctly indicated in the text: ROS stimulate the activity and expression of MMPs. This leads to degardation of collagen and fianlly wrinkles. Rhe cell type that is mainly affected by that are fibroblasts. Why were keratinocytes used for this study?

Reply:

  • Thank you for your interesting question.
  • Firstly, fibroblasts produce collagen. MMPs break down collagen.
  • There are 30 types of MMPs, Morin shows inhibitory effect on MMP-1 expression. MMP-1 is secreted by both keratinocytes and fibroblasts but mainly by keratinocytes.
  • Additionally, PM can penetrate to stratum spinosum – the fourth layer of epidermis, but hard to go deeper down to dermis to induce fibroblasts not synthesizing collagen or secreting more MMPs.

3) Statistics at most instances is absent. This has to be corrected. Each figure with qunatitaive data needs statistsics. the same is ture for all tables.

Reply:

  • Thank you for your suggestion. Statistical analysis has been supplemented to figures and tables.

4) The image quality (resolution) at many instances is horrible and the tetx is very often close to be unreadable.

Reply:

  • Figures are replaced by those with high resolution
  •  

5) Most of the time is good, but has to be improved at many instances. A good example ist the sentecne in line 60 to 64.

Reply:

  • Writing is reviewed and modified to make meaning clearer.
  • For example, in line 59-64: “These deformable liposomes are also known as transferosomes. They are mainly composed of phospholipids and surfactants. In this study, hydrogenated lecithin and tween-80 were used. Tween-80 acts as an edge activator which destabilizes lipid shells, then increases elasticity of the vesicles. This not only improves skin deposition of drug achieved by property of traditional liposomes, but also promotes deeper skin penetration [11]”

6) The pharse "on the other hand" is used very often. the corerct usage would be "on the one hand....on the other hand". Both are linked.

Reply:

  • This is a good suggestion, proper English was used instead of “on the other hand”.

7) Species names are not always italic.

Reply:

  • Thank you for your suggestion. All species names are revised. (Page 1, Line 43-44, Page 2, Line46)

8) Formulas in line 142 and line 250 are basic math and should be removed.

Reply:

  • Thank you for your suggestion. Basic math formulas were removed.

9) I know that some measurements lead to survival rates above 100%. It is somehow okwards that the 110% survival (that is impossible) is especailly mentioned in the text (it is fine for the figure) (line 315).

Reply:

  • I totally agree with you. Line 310-312 were rewritten: “In this experiment, MTT assay illustrated that almost 100 % of HaCaT cells maintained their usual metabolization and proliferation when treated with morin at concentration 10, 25, 40 and 100 µM.”

10) line 318: What doess abnormally grew mean?

Reply:

  • Line 313-314 were modified: “In contrast, survival rate decreased by nearly 30% in group exposed to 200 µM of morin for 24 h.”

11) line 325: What does "induced with PM by increasing ROS" mean?

Reply:

  • I did not make myself clear. Line 322-323 should be rewritten: “PM increased ROS level in HaCaT cells by nearly two and a half folds when compared to negative control group.”
  •  

12) explain + and # in all figures when used. I assume * means normalized to control without PM, # to PM without morin????

Reply:

  • +, - mean with or without exposing to PM or morin/ LMF7, respectively.
  • *, # indicate statistically significant difference with control group (without treating PM and morin) and PM group (only treating PM), respectively, p < 0.05. Explanation was added in all figures.

13) The expmnation of the WB in the text is WRONG: line 338ff: It is not a downregualtion of ERK and p38 but a decraesed phosphorlyation. Or in other words: the english at that section should be corrected/explained in more detail

Reply:

  • Thank you for your suggestion. I was a bit confused of this, but it is clear now. The antibodies used in this study should be phosphorylated ERK (p-ERK) and phosphorylated p38 (p-p38)
  • Line 333-337 were rewritten: “As shown in Figure 4, PM obviously increased the levels of p-ERK and p-p38 in HaCaT keratinocytes by nearly three times. However, in cells first exposed to PM and then treated with morin, their p-ERK and p-p38 expression significantly decreased. 25 and 40 µM of morin were effective in reducing the expression of p-ERK and p-p38 in HaCaT keratinocytes induced by PM.”

14) Table 2: The same as in 13. It ist not only protein expression but also the phosphorylation status. To celarly distinguish two antibodies should have been used (one for ERK, one for p-ERK).

Reply:

  • Thank you for your suggestion. Related text is also revised and modified

15) The quality of Figure 6 should be improved:  Same distances between images itself and between images and text

Reply:

  • Modification has been made on figure 6.

16) Eventually the effect of polydiversity on the interperation of the results should be explained more deeply. The abbreavation P.I should be explained. I assume P.I index is wrong (line 410). It should be either P. index or only P.I.

Reply:

  • I. explanation was supplemented. Line 409-415: “Besides, size distribution is an important characteristic of liposomal vesicles and could be examined by P.I.. High value of P.I. which is close to 1 indicates a broad variation in size or presence of vesicular aggregation. It can be seen that formulation 3 produced the most uniform liposomes due to the smallest P.I. (0.270 ± 0.026). In contrast, a wide range of particle size with P.I. 0.656 ± 0.056 was in formulation 8. Other formulations including 1, 2 and 7 resulted acceptable values of P.I. for lipid-based carriers, which were 0.354, 0.355 and 0.389, respectively.”

17) There is some discrepancy between the absorption test and the biological assays. The abosrption ets was performed on(/in dermis, whereas all other assay aimed on epidermis/keratinocytes. That should be commented.

Reply:

  • In vitro dermal absorption/ percutaneous penetration test was based on the guideline of the European Cosmetic Toiletry and Perfumery Association (CO-LIPA).
  • Dermal absorption is considered as amount of substance found in viable epidermis (epidermis without stratum corneum) and dermis.
  • According to absorption test in this study, the majority of liposomal morin was deposited in the viable epidermis. It is therefore reasonable to investigate bioactivities of morin/ liposomal morin in keratinocytes

Reviewer 2 Report

The manuscript "Liposomes encapsulating morin: investigation on physicochemical properties, dermal absorption improvement and anti-aging activity in PM-induced keratinocytes" presents a systematic study on the encapsulation of morin into liposomal vesicles, obtained using different formulations, and their potential in dermocosmetics. The study is interesting and well designed. The experimental methodologies used and the results obtained are clearly described. In my opinion it can accepted for pubblication after minor revision. The meaning of PM is known to most ot the readers but it is better to give its definition in the abstract. Line 80-81 along with "therefore" at line 82 may be deleted. Moreover it is not clear why some tests were done only on F1 and F7 samples. Please explain it.

Author Response

Dear Reviewer,

 Thank you for your thorough comments and useful suggestion. Below is the author’s response and we hope that it could clarify your remarks 

Best Regards

Yen FL

The manuscript "Liposomes encapsulating morin: investigation on physicochemical properties, dermal absorption improvement and anti-aging activity in PM-induced keratinocytes" presents a systematic study on the encapsulation of morin into liposomal vesicles, obtained using different formulations, and their potential in dermocosmetics. The study is interesting and well designed. The experimental methodologies used and the results obtained are clearly described. In my opinion it can accepted for pubblication after minor revision. The meaning of PM is known to most ot the readers but it is better to give its definition in the abstract. Line 80-81 along with "therefore" at line 82 may be deleted. Moreover, it is not clear why some tests were done only on F1 and F7 samples. Please explain it.

Reply:

  • Thank you for your suggestion. PM is an abbreviation and we added full form of it in the abstract. More details about PM are provided in the introduction section.
  • “Therefore” was removed in line 82.
  • According to results of morphological measurement by TEM and DLS, F1 and F7 produced liposomal morin with the smallest particle size. Small diameter of liposomes is desirable for topical preparation since it is expected to provide better penetration through epidermis. In addition, we also consider proportion of lecinolws-50 and tween 80 in the formulation. High portion of tween 80 in formulation 4 (1 : 1 : 20 : 5), 5 (1 : 5 : 20 : 5), 6 (1 : 10 : 20 : 5) would produce more micelles with single-layer structure rather than liposomes with bilayer shells. Formulation with more micelles could display better water solubility (Table 3) but it is out of this study perspective. Formulation 2 (1 : 40 : 20 : 5) and 3 (1 : 80 : 20 : 5) with larger amount of lecinolws-50 are assumed to produce more rigid lipid shells that could restrict skin penetration. Lastly, inclusion of lecinolws-50 and tween 80 in high ratio led to significant cell death even at low concentration, we tested this on HaCaT keratinocytes. For those reasons, F1 and F7 were used for intermolecular bonding and in vitro dermal absorption test. F7 with better dermal absorption would then be used for DPPH, ROS and Western blotting assay.

Round 2

Reviewer 1 Report

In the current form the paper is now acceptable for publication

Author Response

Dear Reviewer,

We are indeed very glad that you consider this paper for publication.

We would like to sincerely thank you for all of your comments and suggestions. We greatly appreciate the time and effort you take to review the manuscript and give us valuable recommendations. Your expertise helped us a lot in improving the quality of the article.

Yours sincerely,

Prof. Yen FL